# Near-field thermophotovoltaics for efficient heat to electricity conversion at high power density

Rohith Mittapally [1,7], Byungjun Lee [2,7], Linxiao Zhu[3], Amin Reihani[1], Ju Won Lim [4], Dejiu Fan [2], Stephen R. Forrest [2,4,5 ✉], Pramod Reddy [1,2,4 ✉] & Edgar Meyhofer [1,6 ✉]

Thermophotovoltaic approaches that take advantage of near-field evanescent modes are being actively explored due to their potential for high-power density and high-efficiency energy conversion. However, progress towards functional near-field thermophotovoltaic devices has been limited by challenges in creating thermally robust planar emitters and photovoltaic cells designed for near-field thermal radiation. Here, we demonstrate record power densities of ~5 kW/m² at an efficiency of 6.8%, where the efficiency of the system is defined as the ratio of the electrical power output of the PV cell to the radiative heat transfer from the emitter to the PV cell. This was accomplished by developing novel emitter devices that can sustain temperatures as high as 1270 K and positioning them into the near-field (<100 nm) of custom-fabricated InGaAs-based thin film photovoltaic cells. In addition to demonstrating efficient heat-to-electricity conversion at high power density, we report the performance of thermophotovoltaic devices across a range of emitter temperatures (~800 K–1270 K) and gap sizes (70 nm–7 μm). The methods and insights achieved in this work represent a critical step towards understanding the fundamental principles of harvesting thermal energy in the near-field.

[1] Department of Mechanical Engineering, University of Michigan, Ann Arbor, MI, USA. [2] Department of Electrical Engineering and Computer Science, University of Michigan, Ann Arbor, MI, USA. [3] Department of Mechanical Engineering, The Pennsylvania State University, University Park, PA, USA. [4] Department of Materials Science and Engineering, University of Michigan, Ann Arbor, MI, USA. [5] Department of Physics, University of Michigan, Ann Arbor, MI, USA. [6] Department of Biomedical Engineering, University of Michigan, Ann Arbor, MI, USA. [7] These authors contributed equally: Rohith Mittapally, Byungjun Lee. ✉email: stevefor@umich.edu; pramodr@umich.edu; meyhofer@umich.edu

D irect conversion of heat to electricity is expected to play a critical role in developing novel thermal energy storage and conversion[1] technologies. Thermophotovoltaic (TPV) devices that are composed of a hot thermal emitter and a photovoltaic (PV) cell are currently being actively explored for such energy-conversion applications. In TPV devices, electromagnetic radiation emitted by a hot body, when incident on a PV cell, generates electrical power via the photovoltaic effect (see reviews[2,3]). The performance of a TPV system is characterized by two metrics: efficiency, which is defined as the ratio of electrical power output to the total radiative heat transfer from the hot emitter to the PV cell at room (or ambient) temperature, and the power density that is the electrical power output per unit area. Recently, efficiencies of up to 30% in the far field have been reported[4,5], where the emitter (at ~1450 K) and the PV cell are separated by distances larger than the characteristic thermal wavelength. However, the power densities of far field TPV systems are constrained by the Stefan–Boltzmann limit, since only propagating modes contribute to energy transfer. This limit can be overcome by placing the hot emitter in close proximity (nanoscale gaps) to the PV cell, where, in addition to the propagating modes, evanescent modes also contribute and dominate the energy transfer. The enhancements in heat transfer via near-field (NF) effects have long been predicted[6–8] and directly demonstrated in recent work[9–13], paving the way for TPV applications. In fact, several computational studies[14–27] have suggested that it is possible to achieve high-power, high-efficiency TPV energy conversion via NF effects.

In spite of these predictions, few experiments have probed NFTPV energy conversion. This limited progress is due to multiple challenges associated with creating thermal emitters that are robust at high temperatures, creating high-quality PV cells for selectively absorbing above-band-gap NF thermal radiation and maintaining parallelization while precisely controlling the gap between the heated emitter and the PV cell. Recently, a NFTPV system developed by some of us (using a Si emitter and an InAs cell) demonstrated significant enhancements in power output compared with the far field[28] but featured very low efficiencies (<0.1%) and low-power output (~6 W/m²). Further, two other experiments also reported large enhancements in power output compared with the far field by employing different experimental platforms[29,30]. Nevertheless, all of these demonstrations show limited efficiency and power density, with the best-reported device[29] (using a Si emitter and an InGaAs cell) featuring a maximum efficiency of ~0.98% at a power density of ~120 W/m² when operated at a maximum temperature of 1040 K. More recently, another work[31] probed the principles of NFTPV energy conversion in a sphere–plane geometry using a spherical graphite emitter and InSb PV cells that were cryogenically cooled to obtain high cell efficiency. However, given the significant energy expenditure in cooling such devices, the overall efficiency is expected to be low. Thus, high-performance NFTPV demonstrations were limited due to emitters operating at relatively low temperatures and PV cells with poor performance.

## Results

### Devices and experimental setup for exploring efficient NFTPV energy conversion.

To explore the principles of high-efficiency NFTPV energy conversion from planar surfaces and PV cells operating at room temperature, we developed microdevices capable of being heated to temperatures as high as 1270 K, along with matching thin-film PV cells (Fig. 1a) with a spectral response that is capable of absorbing above-band-gap (ABG) thermal radiation while minimizing absorption of sub-band-gap (SBG) photons[19,21,22]. To elaborate, the emitter features a monolithic, doped silicon cantilever with a circular mesa (see Fig. 1b and "Methods" for details) connected to a substrate at room temperature by two stiff beams (Fig. 1a & 1d). The two beams form an electrical resistor ($R_{emitter}$) that can be employed to elevate the temperature of the mesa ($T_{emitter}$) by distributed Joule heating ($j^2\rho$), where $j$ and $\rho$ are the local current density and resistivity, respectively. Also, a 10-nm-thick layer of AlN (Fig. 1b) was conformally deposited over the emitter to form both an electrically insulating layer and a diffusion barrier to protect the emitter surface from degrading at high temperatures[32].

The PV cell has a circular active area of diameter 190 μm (Fig. 1c) to closely match the dimensions of the emitter, and features a thin-film $In_{0.53}Ga_{0.47}As$ (InGaAs) layer epitaxially grown by solid source molecular beam epitaxy on an InP wafer, and transferred to a silicon substrate (see Methods). The top and bottom Au layers serve as electrical contacts (Fig. 1e). The bottom contact also acts as a back surface reflector (BSR) for recycling SBG photons back to the emitter[4,5]. The emitter and the PV cell, as verified by dark-field optical microscopy[33] and AFM scans of the mesa (Fig. 1f) and active area (Fig. 1g), are extremely flat and free of particles and other contamination that would interfere with the NF operation of the TPV system.

To parallelize the emitter and the PV cell (see Methods for parallelization procedure), we employed a nanopositioning platform[12,28,33,34] in a high-vacuum environment (~1 μTorr), and varied the gap size between the emitter and the PV cell from micrometers to nanometers even while the emitter was heated to high temperatures (Fig. 1a). This was accomplished by applying a bipolar voltage across the two terminals of the emitter and maintaining the voltage of the mesa close to the ground potential (see Supplementary Note 4), thus reducing electrostatic interactions with the PV cell, and enabling creation of small gap sizes. Further, no additional active thermal management (i.e., refrigeration) was applied to the PV cell, as the heat transfer is primarily localized to the mesa region of the emitter interacting with the PV cell (see Supplementary Note 5).

### Experimental scheme for probing NFTPV energy conversion.

Here we describe the experimental strategy for heating the emitter, controlling the gap between the parallelized devices, and measuring the power output at each gap size. We began our experiments by passing a current of ~70 mA through the two terminals of the emitter (Fig. 1a). This results in a power dissipation of $P_{Joule} = 411.8$ mW within the beams of the emitter and heats the mesa to a temperature, $T_{emitter} = 930$ K, as determined by a scanning thermal probe-based method (Supplementary Note 6 and ref. [35]). The heated emitter and PV cell were placed at an initial separation of ~7 μm using a coarse-positioning stepper motor that controls the position of the PV cell. The PV cell was then stepped closer to the emitter using a feedback-controlled piezoelectric actuator. The data corresponding to this process are shown in Fig. 2a, where the top panel shows that large steps of ~800 nm are taken initially followed by finer steps of ~2 nm before contact. The electrical resistance ($R_{emitter}$) of the emitter (third panel, Fig. 2a) and the short-circuit current ($I_{sc}$ at $V = 0$) measured across the PV cell (schematic, fourth panel Fig. 2a) at each gap size do not change significantly during the initial steps, but a large variation is seen over the last hundreds of nanometers due to NF enhancement. A sudden jump in the optical signal that monitors deflection of the emitter, which is accompanied by a simultaneous change of $R_{emitter}$ and $I_{sc}$, at the end of the approach, clearly indicates contact (see Methods) between the devices. At this point, the PV cell is quickly withdrawn, to separate the devices back to the initial gap of 7 μm.

To measure the electrical power output of the PV cell, its current–voltage (I–V) characteristics are measured at each gap

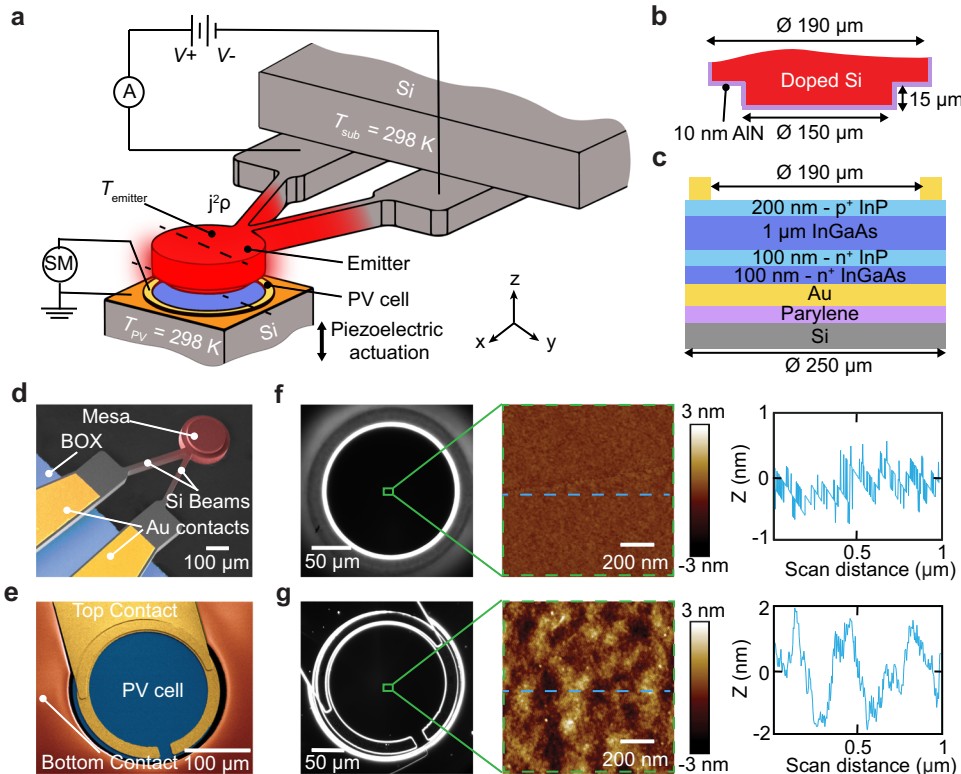

**Fig. 1 Devices and experimental setup. a** Schematic depiction of the experimental setup employed for near-field thermophotovoltaic measurements. The custom-fabricated Si emitter features a suspended mesa (see panel **d**) that is Joule heated (heat dissipation quantified with an ammeter 'A') up to 1270 K by applying a bipolar voltage ($V+$, $V-$) to the two beams. The epitaxially grown InGaAs photovoltaic (PV) cell is moved towards the emitter via a piezoelectric actuator to systematically control the gap size while the electrical power generated is quantified with a source meter (SM). The emitter substrate and the PV cell are at a temperature of ~298 K. **b, c**, Cross-sectional profiles of the emitter and the PV cell at the sections along the black dashed lines in Fig. 1a. **d**, False-colored scanning electron micrograph (SEM) of the emitter with mesa, showing the buried oxide layer (BOX) and the gold contacts on the Si beams. The Si beams featuring a temperature gradient are depicted in red. **e**, False-colored SEM of the PV cell showing the central active layer of the PV cell (blue) as well as top (yellow) and bottom (orange) Au contacts. **f, g**, Dark-field microscope (left panels), atomic-force microscopy (AFM) images (middle panels), and surface roughness profiles (corresponding to the blue dashed lines in the AFM images) of the mesa (f) and the PV cell (g) are shown in the right panels. The peak–peak roughness of the mesa is ~1 nm, while that of the PV cell's active surface is ~4 nm.

size (see Methods). Typical curves are shown for gaps of 7 µm, 200 nm and 100 nm in Fig. 2b, where a clear upward shift of the $I$–$V$ curve to larger short-circuit currents ($I_{sc}$) and moderately increased open-circuit voltage ($V_{oc}$) is seen with decreasing gap size. The increase in $I_{sc}$ from 9.8 µA at 7 µm, to 56 µA at 100 nm, can be attributed to the increased above-band-gap (ABG) photon flux from evanescent modes coupled at sub-wavelength gaps (see below). The electrical power output at the maximum power point ($P_{MPP}$, Fig. 2c) of the $I$–$V$ curve is $P_{MPP} = FF \times V_{oc} \times I_{sc}$, where $FF$ is the fill factor (at 100 nm, $FF = 0.73$). The variation of $P_{MPP}$ with gap size is plotted in Fig. 2d (violet squares, left axis), where the PV cell power output remains around 2 µW for gaps from 7 µm to 600 nm. Below 600 nm, the power output increases substantially to 14.8 µW at the smallest gap of 70 ± 2 nm, indicating an ~8-fold power enhancement in the NF when compared with the far field. To interpret this NF enhancement, all the surfaces of the emitter that contribute radiative energy fluxes to the PV cell must be considered. The surfaces of the emitter are labeled 'mesa' and 'rec' (see schematic Fig. 2d), where 'mesa' refers to the central region ($A_{mesa} = 7.07 \times 10^{-8}$ m²) and 'rec' signifies the recessed ring ($A_{rec} = 4.2 \times 10^{-8}$ m²) surrounding the mesa. When considering only the contribution from the $A_{mesa}$, the NF power enhancement is 11-fold relative to power generation in the far field, whereas a smaller 8-fold enhancement is observed when contributions from $A_{rec}$ are included in the power transfer as seen in the experimental data of Fig. 2d. This is

because only the mesa enters the NF of the PV cell, while $A_{rec}$ always remains in the far field. Thus, the actual enhancement can be larger if all surfaces are brought into the NF.

To understand the physical mechanisms behind the enhancement, we developed a model based on the formalism of fluctuational electrodynamics[7]. Specifically, we employed our model (Methods, Supplementary Note 7) to estimate the power output $P_{MPP}$ and the total radiative heat transfer $Q_{RHT}$ as functions of $T_{emitter}$ and gap size for the geometries (including $A_{mesa}$ and $A_{rec}$) and materials that correspond to those employed in this work. The estimated $P_{MPP}$ is plotted as a purple line in Fig. 2d, which agrees with the experimentally measured $P_{MPP}$. Further, the calculated $Q_{RHT}$ is observed to continuously increase from ~72 µW at 7 µm, to ~1 mW at 70 nm.

**NFTPV performance at temperatures above 1000 K.** To understand the temperature-dependent performance of the TPV system, we systematically increased $T_{emitter}$ in steps of ~100 K and performed experiments as described above. When the emitter temperature increases, the characteristic wavelength of the radiated spectrum decreases, increasing the fraction of energy in the ABG region, and correspondingly the photocurrent ($I_{sc}$). As the emitter temperature is raised from 1050 K to 1270 K, in Fig. 3a, we observe that $I_{sc}$ increases from 30 µA to 150 µA. Importantly, a large shift in $I_{sc}$ is seen as the gap size is reduced from 7 µm to ~100 nm, for example, at the highest temperature of 1270 K, an

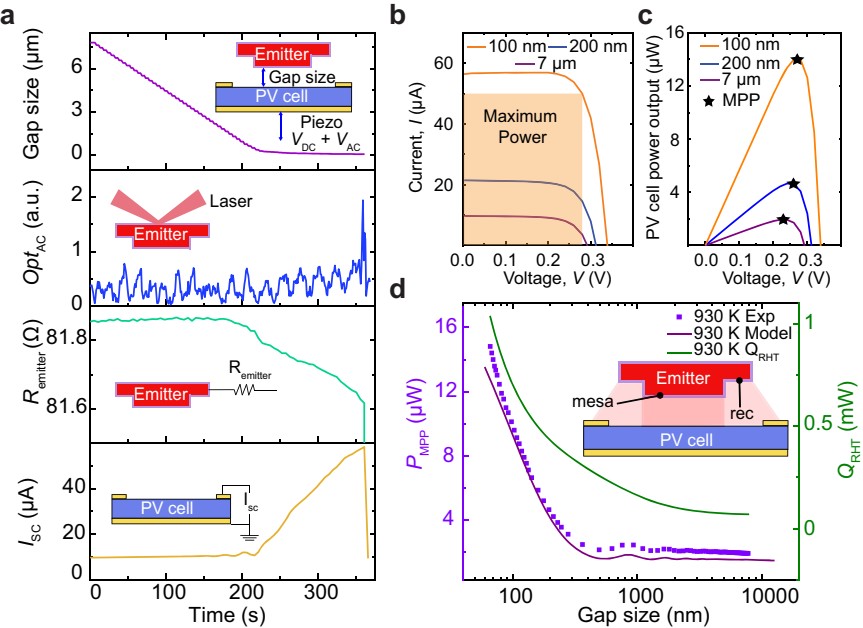

**Fig. 2 Experimental data with an emitter heated to 930 K and the photovoltaic cell at room temperature. a** Data obtained as the gap size between the emitter and the photovoltaic (PV) cell is reduced from ~7 μm to contact. The top panel shows the gap size as a function of time. The inset shows how the PV cell mounted on a piezoelectric actuator is displaced ($V_{DC}$ is the DC voltage applied to the feedback-controlled piezo and $V_{AC}$ is a superimposed small AC-modulating signal at 4 kHz). The second panel shows the variation of the AC optical signal ($Opt_{AC}$), while the third and fourth panels show the changes in the resistance of the emitter ($R_{emitter}$) and short-circuit current ($I_{sc}$) in the PV cell. The simultaneous jumps in $Opt_{AC}$, $R_{emitter}$ and $I_{sc}$ indicate contact. **b** Data from current–voltage (I–V) characterization performed for gaps of 7 μm, 200 nm and 100 nm. The orange-shaded region represents the maximum power extractable from the PV cell for a gap size of 100 nm. **c** The power output of the PV cell as a function of the voltage showing the maximum power points (MPP). **d** The power output at the maximum power point ($P_{MPP}$) at different gaps. Violet squares indicate the measured data, the purple solid line represents the theoretically estimated $P_{MPP}$, and the green line shows the total radiative heat transfer ($Q_{RHT}$) between the emitter and the PV cell as a function of gap size. The inset (not to scale) indicates that both the 'mesa' (near-field contribution from the circular region in the center) and 'rec' (far field contribution from the 15-μm-recessed circular ring) surfaces contribute to the total $P_{MPP}$ and $Q_{RHT}$.

~5-fold increase in $I_{sc}$ is measured (purple solid and dashed lines in Fig. 3a). We note that the I–V curve at 1050 K and a gap size of 100 nm is similar in shape to that of one obtained at 1270 K in the far field, highlighting that NFTPVs can achieve similar or higher power outputs at significantly lower temperatures when contrasted to a comparable far field TPV device. Further, in Fig. 3b, we plot $V_{oc}$ as a function of $I_{sc}$ for the different temperatures and gap sizes (the direction of the arrows signifies decreasing gap size), which indicates a logarithmic dependence of $V_{oc}$ on $I_{sc}$, characteristic of PV cells (see Supplementary Note 7). Thus, $V_{oc}$, $I_{sc}$ and $P_{MPP}$ increase with decreasing gap size and increasing temperature. Further, the calculated $V_{oc}$ and $I_{sc}$ (solid lines in Fig. 3b) agree with the experimental data over the broad range of temperatures and gap sizes explored.

The measured $P_{MPP}$ as a function of gap size is plotted in Fig. 3c at different temperatures between 810 and 1270 K. At all temperatures, $P_{MPP}$ increases when the gap size is decreased sufficiently, for example, at 1050 K, the power output increased from ~7 μW at 7 μm, to 41 μW at a 90-nm gap, a six-fold increase due to NF enhancement. The measured (various symbols) $P_{MPP}$ agree well with that estimated from our model (color bands corresponding to $T_{emitter} \pm \Delta T$, where $\Delta T = 27$ K when $T_{emitter} = 1270$ K and $\Delta T = 10$ K for other temperatures, as 10 K is the upper bound to uncertainty in that temperature range). Nonmonotonic changes in the experimental power outputs are seen for gap sizes between 500 nm and 7 μm at all temperatures due to interference effects, highlighting the capability of our platform to resolve such effects in agreement with the model.

The NFTPV energy conversion efficiency ($\eta$), defined as the ratio of the measured power output $P_{MPP}$ to the calculated total

radiative heat transfer $Q_{RHT}$ to the PV cell ($\eta = (\frac{P_{MPP}}{Q_{RHT}}) \times 100$), is plotted in Fig. 3d as a function of gap size and temperature (color bands correspond to efficiencies obtained by calculating $Q_{RHT}$ in a temperature interval of $T_{emitter} \pm \Delta T$, where $\Delta T = 27$ K for $T_{emitter} = 1270$ K and 10 K for other temperatures, as described above). Here, $\eta$ increases with temperature, independent of gap size. For example, at 100-nm gaps, an increase in efficiency from 0.5% to 6.8% is observed as the emitter is heated from 810 K to 1270 K. We note that at temperatures >930 K, the efficiency is greater than the highest efficiencies reported in the literature[28–30]. At a given temperature, the efficiency initially decreases with gap size for the smallest gaps, then starts to increase, as predicted by our model (see below).

To understand the detailed spectral characteristics of NF energy transfer, we calculate the spectral energy transfer (Fig. 4a) from the emitter at 1270 K to the PV cell at 300 K for a range of gap sizes. For example, at a gap size of 100 nm, significant enhancement over the blackbody limit (black dashed line) can be seen in the ABG energy transfer, while considerable suppression of SBG energy transfer below the blackbody limit is observed, due to the incorporation of a thin-film back reflector (see Supplementary Note 10 for comparison with a bulk PV cell). The residual SBG energy transfer has contributions from surface phonon–polaritons in the low-frequency range (~14% of $Q_{RHT}$ in 0.0124–0.073-eV range) while the rest of the absorption primarily occurs in the Au BSR (~55% of $Q_{RHT}$ in 0.074–0.74 eV range). The power generating component of the ABG spectrum absorbed in the active layer ($P_{AL}$) is shaded in orange (~26% of $Q_{RHT}$). Approximately 32% of $P_{AL}$ is extracted as electrical power, while

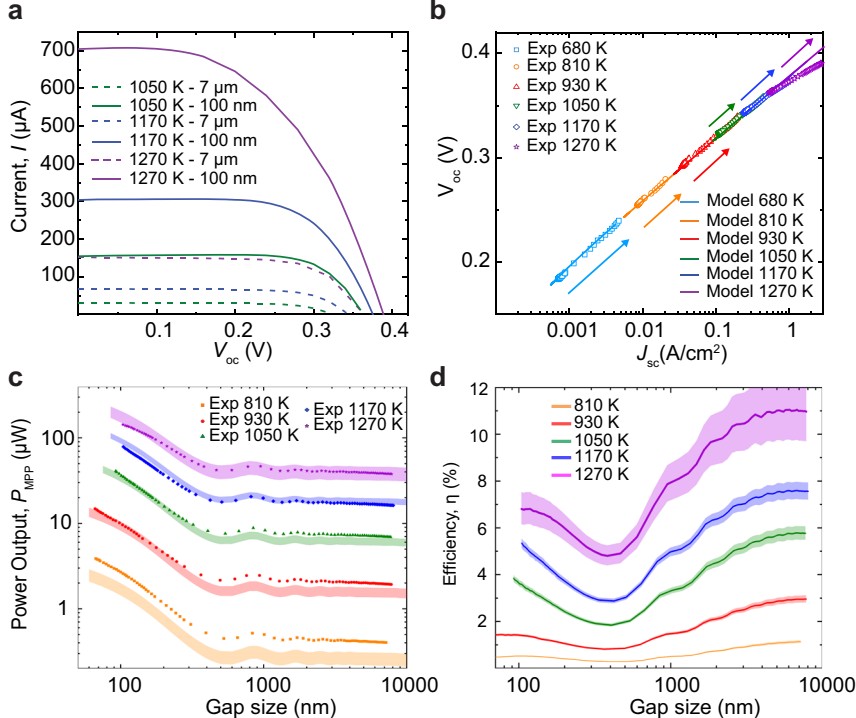

**Fig. 3 Performance of the thermophotovoltaic system as a function of temperature and gap size. a** Current–voltage ($I$–$V$) curves obtained at three emitter temperatures and two gap sizes, indicating an upward shift with increasing temperature and reduced gap size. Dashed lines and solid lines correspond to 7-μm and 100-nm gap sizes, respectively. **b** $V_{oc}$ (open-circuit voltage) as a function of $J_{sc}$ (defined as short-circuit current ($I_{sc}$) per unit area of the PV cell) obtained from experimental data (various symbols) and calculations (solid lines) at all temperatures. The arrow direction indicates decreasing gap size. At high temperatures, there is some overlap in the data sets. **c**, The experimentally measured power output ($P_{MPP}$) at different temperatures as a function of gap size, indicating power enhancements as the gap size is reduced from 7 μm to 100 nm. The shaded regions indicate the theoretical power output ($P_{MPP}$) based on our model in a temperature range of $T_{emitter} \pm \Delta T$, $\Delta T = 27$ K for $T_{emitter} = 1270$ K and 10 K otherwise. **d** The efficiency ($\eta$) defined as the ratio of $P_{MPP}$ to the calculated total radiative heat transfer ($Q_{RHT}$) at different temperatures of the emitter as a function of gap size. The solid lines correspond to the efficiencies obtained by calculating $Q_{RHT}$ at $T_{emitter}$, while the shaded regions correspond to the efficiencies due to the uncertainty ($\Delta T$) in the measurement of $T_{emitter}$.

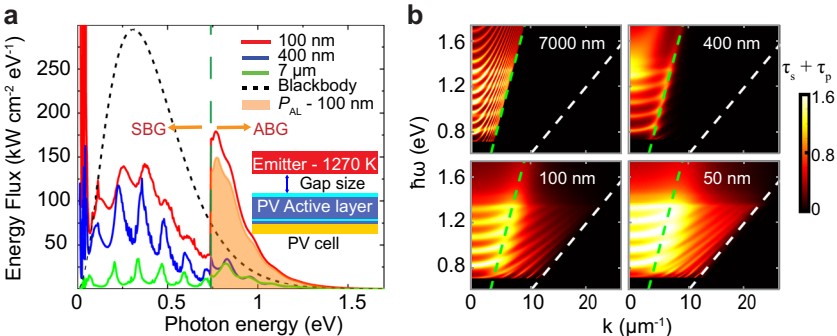

**Fig. 4 Physical mechanism of NF enhancement. a** The spectral energy transfer from a hot thermal emitter at 1270 K to a photovoltaic (PV) cell at 300 K is plotted as a function of photon energy for three gap sizes. The black dashed line represents the blackbody radiative limit between two semi-infinite plates at 1270 K and 300 K. Enhancement in above-band-gap (ABG) transfer is seen as the gap size is reduced down to 100 nm. The orange shaded region represents the radiative energy transfer ($P_{AL}$) from the emitter to the InGaAs active layer, which drives the generation of charge carriers. Green dashed line represents the band gap of the PV cell while SBG represents the sub-band-gap region. Note that the energy flux for the 100-nm gap size at low energies (<0.1 eV) extends beyond the y-axis range. **b** The total transmission function of different modes from the emitter to the active layer, as a function of photon energy and parallel wavevector at four gap sizes. The green dashed line represents the light line in vacuum, while the white line represents the dispersion relation in the top InP layer.

the rest is lost due to thermalization, nonradiative recombination and ohmic losses (see Supplementary Note 7).

Next, the efficiency trend as a function of gap size can be understood by comparing the spectral energy transfer at three gap sizes of 7 μm (far field), 400 nm, and 100 nm. In the far field (green line), a large suppression of SBG energy transfer is observed that is related to the thin-film BSR[4,5]. Even when we reduce the gap size, the SBG energy transfer remains below the blackbody limit. Moreover, as the gap size is reduced from 7 μm to 400 nm, SBG energy transfer is observed to increase more rapidly than ABG energy transfer. These differences in the rates of change of SBG and ABG energy transfer cause an initial drop

in the efficiency in Fig. 3d at intermediate gaps around 500 nm. As the gap size is further reduced to 100 nm, ABG energy transfer exceeds the blackbody limit, whereas a comparatively smaller rise in SBG energy transfer results in the efficiency increase at smaller gaps.

To further elucidate the contribution of different modes to the observed NF enhancement in $P_{AL}$, we evaluate the transmission coefficients of $s$ and $p$-polarization modes ($\tau_s + \tau_p$) as a function of photon energy ($\hbar\omega > 0.75$ eV) and parallel wavevector ($k$) (Fig. 4b). In the far field at a gap of 7 μm, only propagating modes above the light line in vacuum contribute to ABG energy transfer, whereas in the NF at 100 nm, evanescent modes between the light line in vacuum (green dashed lines in Fig. 4b) and in the top substrate of the PV cell (white dashed lines) also contribute, leading to a broadband enhancement in ABG energy transfer.

The performance of a PV cell under illumination is generally determined by the short-circuit current ($I_{sc}$), open-circuit voltage ($V_{oc}$) and the fill factor ($FF$). While $I_{sc}$ depends on the incident photon flux, the internal quantum efficiency and series resistance (weak dependency due to low series resistance) of the PV cell, $V_{oc}$, and $FF$ depend on various factors such as the nonradiative recombination, series and the shunt resistances of the PV cell (see Supplementary Note 9 for dark I–V characteristics of the PV cell and the variation of $FF$). In our experiments, $I_{sc}$ (Fig. 5a) is observed to increase with more-than-linear dependency on temperature at both gap sizes of 100 nm (NF, green circles) and 7 μm (far field, violet squares). Similarly, the variation of $V_{oc}$ with temperature is plotted in Fig. 5b along with the theoretical

calculations. The experimental data agree quite well with the theoretical calculations. Specifically, the agreement in $V_{oc}$ with our model, which does not include temperature dependency of the PV cell, indicates that the cell remained close to room temperature during our measurements.

Finally, the power density and efficiency in the far field (7 μm) and NF (100 nm), respectively, as functions of temperature, are shown in Fig. 5c and d. A clear enhancement in power density is observed at all temperatures (~7× at 810 K and ~4× at 1270 K). The estimated efficiency from our calculations of $P_{MPP}$ and $Q_{RHT}$ is $\eta = \left(\frac{P_{MPP}}{Q_{RHT}}\right) \times 100$ ~8.3% (green dashed line), which is slightly higher than the efficiency estimated from the experimental power output (~6.8%). This ~18% disagreement at the highest temperature with the theoretically predicted value may be attributed to uncertainty in temperature measurement of the emitter, modeling parameters, such as the dielectric properties of the emitter as a function of temperature, and the PV cell's series and shunt resistances or a small increase in the temperature of the PV cell. The efficiencies in the NF are slightly smaller than in the far field, owing to absorption in the Au film reflector, which can be mitigated by engineering the devices to further suppress SBG energy transfer. This can be achieved by employing an air-gap PV cell, which has recently been shown to support very efficient SBG suppression[4]. Such devices must be engineered to address a host of technical requirements (smooth surfaces, planarity, and temperature compatibility) before they can be adapted for NFTPV studies.

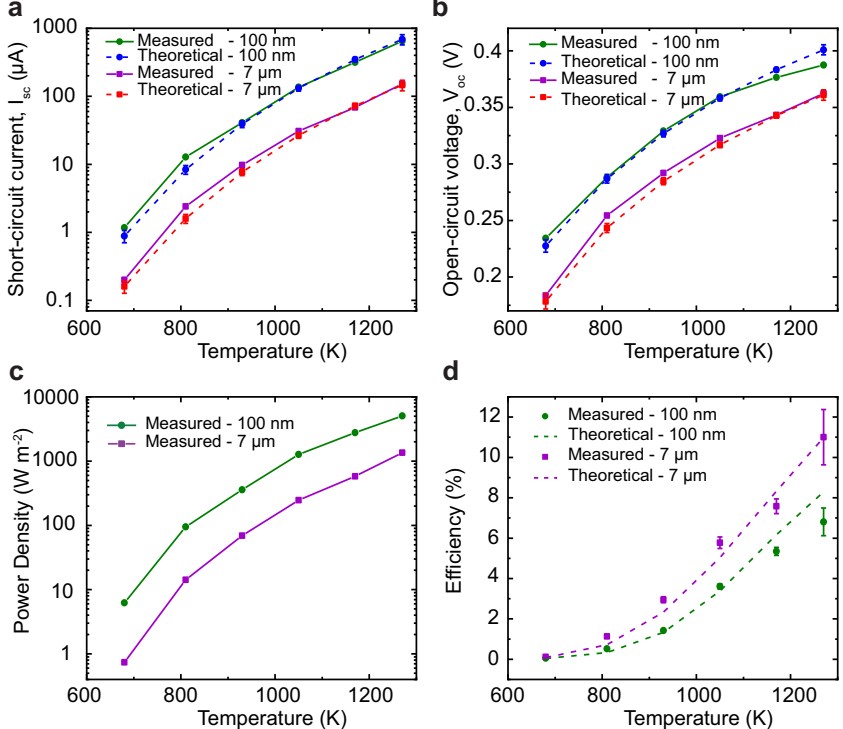

**Fig. 5 Performance of near-field thermophotovoltaic system (NFTPV). a, b** Measured and calculated short-circuit current ($I_{sc}$) and open-circuit voltage ($V_{oc}$) as a function of temperature. Green circles and violet squares represent the experimental data points, while blue circles and red squares represent the calculated data points with the corresponding uncertainties, at gap sizes of 100 nm and 7 μm, respectively. Solid and dashed lines added as a guide to the eye. **c** The measured power density ($P_{MPP}$ per unit area of the PV cell) is plotted as a function of temperature at two gap sizes, one in the far field (violet squares) and other in the near-field (green circles, solid lines plotted as visual guide). **d** Efficiency of NFTPV system at different temperatures for two gap sizes, defined as the ratio of the measured power output ($P_{MPP}$) and theoretical radiative heat transfer ($Q_{RHT}$), calculated at $T_{emitter}$ with an uncertainty of ± $\Delta T$, where $\Delta T = 27$ K for $T_{emitter} = 1270$ K and 10 K otherwise. The dashed lines represent theoretical estimates of the efficiency based on our model.

## Discussion

We demonstrated efficient (~6.8%, excluding the heat losses through conduction and radiation from surfaces not facing the photovoltaic cell) thermophotovoltaic power generation in the NF (< 100-nm gaps) at a power density of ~5 kW/m² when the emitter is heated to 1270 K and the PV cell is at room temperature. This power density for room temperature PV cells is more than an order of magnitude higher than for the best-reported TPV systems in the NF, while also operating at 6-times higher efficiency[29]. By employing heavily doped silicon, we could successfully operate the emitter in the temperature range of 300–1270 K by Joule heating. Further, the emitters presented here, capable of operation at high temperatures (up to 1270 K), present a platform for future work aiming to experimentally explore novel strategies to improve NFTPV performance by engineering thermal emitters[36–40] or PV cells[4,41]. These advances are expected to help establish the principles necessary for the exploitation of a range of NF-based TPV nanotechnologies. Future studies on the long-term stability of the emitters at high temperatures with various protective coatings, under a range of pressures, could enable realization of practical devices.

## Methods

**Device geometry and electrical setup**. The thermal emitter and the PV cell were custom-fabricated using standard microfabrication and MBE techniques (Supplementary Notes 1 & 2). The emitter is a cantilevered structure that features a large planar island suspended from the substrate through two beams. Silicon was chosen as the emitter material due to its ease of microfabrication and compatibility with other thin-film material growth processes such as ITO, TiN, AlN etc. The heavily doped (~$3 \times 10^{19}$ cm$^{-3}$ B-doped) Si emitter allowed us to reliably heat the emitter in a large temperature range. The suspended island consists of a 150-μm-diameter 'mesa' region and a 190-μm-diameter 'rec' region, which is recessed from the mesa to a depth of 15 μm (Fig. 1b). The beams are each 20-μm wide, 270-μm long, and 45-μm deep, resulting in a typical thermal conductance of $G_{beam} = 400$ μW K$^{-1}$, stiffness of ~2 kN m$^{-1}$ (Supplementary Note 3) in the 800–1300-K temperature range, and a resistance of $R_{emitter} = 80$ Ω. The two beams are electrically isolated from the substrate via a buried oxide layer (labeled BOX in Fig. 1d). A bipolar voltage (V) is applied across the two beams using an Agilent E3631A DC power supply and the current was monitored through an Agilent 34401 A multimeter (Fig. 1a). The PV cell comprises of a (100/100/1000/200 nm) $1 \times 10^{18}$ cm$^{-3}$ n$^+$ InGaAs/$1 \times 10^{18}$ cm$^{-3}$ n$^+$ InP/$1 \times 10^{17}$ cm$^{-3}$ n InGaAs ($E_g \sim 0.75$ eV)/$1 \times 10^{18}$ cm$^{-3}$ p$^+$ InP heterostructure, which is epitaxially transferred to a 500-μm-thick Si handle wafer coated with 2 μm of Parylene-C and a 400-nm Au bottom contact (Fig. 1c, see Supplementary Note 2 for fabrication details). The device has a 250-μm diameter of which a 190-μm-diameter region enclosed by a 20-μm circular Au contact is available for measurement (Figs. 1c, 1e). The PV cell sidewalls and the bottom-contact Au surface are coated with 1-μm-thick PI-2555 for insulation. Finally, the I–V characteristics of the PV cell are measured using a Keithley 2401 sourcemeter (SM in Fig. 1a) between the top and the bottom contacts.

**Parallelization of the devices**. Our custom-built nanopositioner allows lateral alignment of the devices with an accuracy of a few micrometers along the x- and y-directions (see Fig. 1a where the directions are shown), and ~6 μrad of angular alignment about both the axes. The parallelization is achieved in a two-step process. First, coarse alignment is accomplished by imaging the chip surface (~1 cm × 1 cm in size) that has the PV cell integrated, using a 50× microscope objective (Zeiss LD EC Epiplan-Neofluar 50 × /0.55 HD) with a shallow depth of focus of 2 μm. The tip/tilt of the PV chip is then manually adjusted, while translating the chip along x and y directions, to bring the whole chip into focus. Thus, the angular deviation of the PV chip is less than ~200 μrad and consequently the deviation from parallelism across the PV cell surface is less than 40 nm. This tip/tilt process is repeated on the emitter (placed at a safe distance above the PV cell) using a goniometer integrated into our nanopositioner, resulting in a similar deviation across the mesa surface. Thus, in the first coarse-positioning step, the devices are parallelized with a deviation of ~80 nm across the surfaces of the devices. The whole assembly is then moved into a vacuum chamber (~1 μTorr). Upon heating the emitter to a desired temperature, the alignment may be impacted by thermal effects. Therefore, we perform a second in situ parallelization step after heating the emitter to high temperatures, by using the integrated goniometer. To perform this step, we take advantage of the fact that energy transfer from the emitter to the PV cell is maximized when the devices are perfectly parallel. Specifically, we first reduce the gap size between the emitter and the PV cell, until contact is made, record the $P_{MPP}$ at the smallest gap size, and withdraw the PV cell by 10 μm. The tip/tilt of the emitter is then adjusted in steps of ~100 μrad and the approach to nanogaps and contact is repeated to maximize the measured $P_{MPP}$. Following this

iterative second alignment procedure, we estimate a maximum deviation from parallelism of ~15 nm across the 150-μm mesa.

**Detecting contact between the emitter and the PV cell**. To detect mechanical contact between the emitter and the PV cell, we employ a scheme similar to the optical scheme used in atomic force microscopes. Specifically, we focus a laser onto the backside of the emitter and collect the reflected laser beam (schematic in panel 2 of Fig. 2a) on a segmented photodiode with two independent detectors. Further, a small AC signal $V_{AC}$ is applied to the piezoactuator that modulates the gap size between the emitter and the PV cell at an amplitude of ~2 nm at 4 kHz. The 4-kHz component of the difference signal of the two segments in the photodiode ($Opt_{AC}$) is continuously measured in a lock-in amplifier (SRS 830). When the PV cell makes physical contact with the emitter, a change in this signal can be noticed indicating contact (see panel 2 of Fig. 2a). In addition, sudden changes in the simultaneously measured $R_{emitter}$ due to rapid cooling through heat conduction to the PV cell enable us to independently detect contact (Fig. 2a).

**Estimation of emitter temperature**. The temperature of the emitter $T_{emitter}$ for various power dissipations ($P_{Joule}$) was measured using an ultra-high-vacuum scanning thermal microscopy (UHV-SThM) technique. The emitter is loaded into the UHV chamber (UHV 750) of an RHK SPM (SPM 1000) and heated by supplying a known power (e.g., 411.8 mW). Subsequently, a SThM probe with an embedded temperature sensor is brought into contact with the hot emitter and the temperature of the probe and the probe–sample thermal contact resistance are measured (see Supplementary Note 6 and ref. [35]), which enable us to directly estimate the temperature of the emitter. This measurement of $T_{emitter}$ is repeated for various values of $P_{Joule}$ from which the temperatures described in Fig. 3c are obtained. The uncertainty of this temperature measurement is shown in supplementary Fig. 6d. Since the uncertainties associated with our measurements are different across temperatures, we use an uncertainty of ±27 K for the highest temperature and an upper bound of ±10 K for all other temperatures in estimating the uncertainty bands in Figs. 3c and 3d.

**Modeling approach for calculating NF radiative energy transfer**. To model the power output and calculate the total radiative energy transfer between the emitter and the PV cell, we first approximate our devices as infinitely extended in the lateral x, y dimensions and multilayered along the z direction (see Fig. 1a for directions). The thermal emission from each layer is generated by fluctuational currents within that material. The correlations of these fluctuational currents are described by fluctuation–dissipation theorem[42,43] and the resulting energy flux in any layer of the structure is calculated using a numerically stable scattering matrix formulation[44]. Using this method, we calculate $Q_{RHT}$ from different layers of the emitter to the PV cell. To estimate the $P_{MPP}$, we first calculate the spectral photon flux from the emitter to the active layer of the PV cell. The photocurrent generated from this photon flux is incorporated into an equation describing the PV cell and the maximum power $P_{MPP}$ is obtained from the corresponding I–V characteristics. A detailed description of this model can be found in Supplementary Note 7.

**Reporting summary**. Further information on experimental design is available in the Nature Research Reporting Summary linked to this paper.

## Data availability

The data that support the findings of this study are available from the corresponding authors on reasonable request.

## Code availability

The code used to analyze the radiative energy transfer is available from the corresponding authors on reasonable request.

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

## Acknowledgements

S.R.F., P.R., and E.M. acknowledge support from the Army Research Office award no. MURI W911NF-19-1-0279. We acknowledge the Lurie Nanofabrication Facility for facilitating the fabrication of devices.

## Author contributions

S.R.F., P.R., and E.M., conceived the work. R.M. and J.W.L. fabricated the emitter devices. B.L. and D.F. fabricated the PV cells. R.M. conducted the experiments. L.Z. performed the calculations. A.R. performed the temperature characterization. All the authors contributed to data analysis. The paper was written by R.M., S.R.F., P.R. and E.M. with comments and input from all the authors.

## Competing interests

The authors declare no competing interests.
