## [Peer Review File · Nature Communications]

REVIEWER COMMENTS

Reviewer #1 (Remarks to the Author):

In this paper, the authors demonstrated near-field thermophotovoltaic (NTPV) devices with record power densities of $\sim 5\text{kW}$ by using their custom-built nanopositioning platform. Comparing to their pioneering work [Nat. Nanotechnol. 13, 806 (2018)], the authors made several technological advances in their experiment, e.g. they realized the emitter temperature as high as 1270 K and developed InGaAs-based thin film PV cells with high fill factors. However, the basic concept is the same as that of the previous studies; the quantitative evaluation of the NTPV devices across a wide range of gap sizes was already investigated in the above pioneering work, and NTPV devices combining high-temperature ($>1000\text{ K}$) emitters and InGaAs-based thin film PV cells were also demonstrated in Ref. 29. These facts reduce the novelty of this work, and thus it is difficult to judge whether this paper is truly appropriate for Nature Communications or not.

Besides the above overall comment, this referee recommends the authors to address the following issues to improve scientific accuracy.

1. In general, the efficiency of a TPV system is defined as the ratio of the electrical power output to the total input power necessary for heating the emitter. On the other hand, in this paper, the authors defined the efficiency as the ratio of the electrical power output to the calculated total radiative heat transfer from the emitter to the PV cell, which neglected the graybody thermal radiation to the opposite side of the PV cell and the thermal conduction loss through the cantilever. If these losses are taken into account, the efficiency of the demonstrated device will be very low. Therefore, the authors should explicitly explain that there are two definitions of the efficiency and should calculate the efficiency of their devices according to both definitions. They should also modify the abstract and conclusion to improve scientific accuracy, for example as follows: "Here we demonstrate record power densities of $\sim 5\text{kW}/\text{m}^2$ at an estimated efficiency of 6.8 % when neglecting the far-field thermal radiation to the opposite side and the thermal conduction loss."

2. During the near-field experiment, did the authors perform the real-time measurement of the emitter temperature as described in Supplementary Note 6? Or, did they measure the relationship between the dissipated power and the emitter temperature (Supplementary Fig. 6) before the near-field experiment? In the latter case, was the relationship reproduced over repeated measurements, especially for high temperatures? By the way, it is counterintuitive that the temperature of 414.8°C was obtained with the heating power of 0.34 W while almost double temperature (794.1°C) was obtained with the additional heating of only 0.11 W. Why was the temperature increase of the emitter so superlinear?

3. Related to the above comment, the upper bound of the uncertainty of the temperature measurement might be much larger than that claimed in the manuscript (10K). In the arXiv preprint of the authors (Ref. 5 in Supplementary Information), they performed the temperature calibration only at relatively low temperatures. Can the authors discuss the possible uncertainty of the temperature measurement during the near-field experiment at higher temperatures?

4. In Fig. 3, the authors show the calculated results of the electrical power and efficiency, but these values depend on not only the spectral energy transfer but also the performance of the PV cell (i.e. the open-circuit voltage and filling factor). In contrast, the short-circuit current does not depend on the latter if the quantum efficiency of the PV cell is constant. This referee recommends the authors to include the direct comparison of the measured and calculated short-circuit currents.

Reviewer #2 (Remarks to the Author):

In this work, the authors demonstrated a near-field TPV device that shows increased power density and efficiency compared to the one some of them showed previously. This is mainly achieved by using a high-temperature doped Si emitter that has more desirable selective emission spectrum than a broadband emitter. The work shown here is novel and also has potential significance for practical applications in thermal energy conversion. As such, I think this work is suitable for publication. In the meantime, I think the following questions/points need to be addressed:

1) About the emitter. The authors used doped Si as the emitter, presumably because it is suitable for the microfabrication process used here and maybe it is stable at high temperature? However, doped Si might not be the best emitter material if the goal is to achieve narrowband emission optimized for TPV. Can the authors elaborate the reasoning behind using the doped Si (and also the doping concentration chosen here) and comment on other possible materials. Also, what about the temperature stability of this emitter? The measurement is done in UHV. Can the authors comment on the stability of the emitter if a more practical vacuum level (with higher pressure) was used.

2) About the heat loss. The authors determined the efficiency using the measured power output from the PV device and the calculated radiative heat transfer flux, which is relevant for this device geometry. Can the authors also discuss the overall energy efficiency (electrical output divided by electrical power input or joule heating applied on the emitter). More specifically, I think it will be useful to quantify the heat loss (in terms of the percentages) through different pathways. One large heat loss I could imagine is through the side walls of the Si emitter (Si is much thicker compared to the near field gap).

3) About the SBG photons. The authors mentioned that the SBG energy transfer is enhanced with the near field. I guess this is related to the RHT flux (as shown in fig. 4a). This leads to the drop in the efficiency in the near field. However, isn't the TPV cell capable of reflecting (and thus recycling) the SBG photons? So the flux of the SBG photons is presumably not important here. I guess this is the key idea behind refs. 4 & 5 (for far field TPV). Likewise and related to point 2 above, I am not sure if the efficiency calculation should include the RHT flux of the SBG photons. If these photons are recycled, then it should not be included in the calculation (as done in ref. 4). If they are not efficiently recycled, then perhaps the authors should explain this in detail.

Reviewer #3 (Remarks to the Author):

This research is focusing on energy conversion from thermal energy to electricity using near-field radiation transfer. The nanometer-sized parallel gap between a doped-Si emitter and an InGaAs thermophotovoltaic (TPV) cell surfaces is rigorously and successfully established in a vacuum chamber using the smallest effective area to confirm the view factor of almost unity. The output power was enhanced by near-field effect by a factor of 8, compared with that by far-field radiation transfer. Also, numerical results agree well with the experimental ones. As a result, the manuscript is acceptable for publication.

However, Reviewer would like to give some comments as follows.

(1) The author estimated the conversion efficiency from the measured output power and the calculated input energy from the emitter to the TPV cell. However, in Figure 3d, each symbol indicates 'Exp' which seems a conversion efficiency obtained from only experiment. Reviewer suggests that those symbol data should be removed. Generally speaking, the conversion efficiency should be defined using the input energy measured by experiment and the output power obtained by experiment. As the authors know well, it is not easy to measure the input energy experimentally. All researchers agree with you. In your research, fortunately, the calculated output power agrees well with the measured one. If so, the author had better show the conversion efficiency obtained from only calculation which are the shaded region in Fig.3d. Otherwise, it is not fair to make a comparison between your results and previous results obtained only from experiments.

(2) In order to keep the TPV cell temperature at 298 K, the author should set up a cooling system at the bottom of the Si block substrate. The reader can't confirm how to make the thermal management from Figure 1a. In addition, in the Supplementary Figure 5a, the bottom of the Si block substrate is kept at constant (maybe at 298K) in numerical simulation. The volume of the Si block (400 x 400 x 200 microns) is very small; as a result, the heat capacity is also very small. It means if the authors could not prepare the cooling system at the bottom of the Si block, the temperature of cell with Si block would rise up easily by an input heat of 2mW within a short time. The author should mention clearly how to keep the cell temperature at almost room temperature for all experiments. Is there any cooling system or another large substrate contacted with the Si block?

RESPONSE LETTER

We thank the referees for taking the time to review our manuscript and for providing their feedback. We are also grateful for many insightful comments that helped us to improve the manuscript. Below, we provide a detailed point-by-point response to each of the comments of the referees and explicitly point out the changes that we have made to the manuscript and the SI. In order to facilitate reading, we quote the referees' comments in italic blue.

Reviewer #1 (Remarks to the Author):

In this paper, the authors demonstrated near-field thermophotovoltaic (NTPV) devices with record power densities of $\sim 5\text{kW}$ by using their custom-built nanopositioning platform. Comparing to their pioneering work [Nat. Nanotechnol. 13, 806 (2018)], the authors made several technological advances in their experiment, e.g. they realized the emitter temperature as high as 1270 K and developed InGaAs-based thin film PV cells with high fill factors. However, the basic concept is the same as that of the previous studies; the quantitative evaluation of the NTPV devices across a wide range of gap sizes was already investigated in the above pioneering work, and NTPV devices combining high-temperature ($>1000\text{ K}$) emitters and InGaAs-based thin film PV cells were also demonstrated in Ref. 29. These facts reduce the novelty of this work, and thus it is difficult to judge whether this paper is truly appropriate for Nature Communications or not.

As we described in significant detail in our manuscript, progress towards practical near-field thermophotovoltaic devices has been severely limited by the lack of thermally robust planar emitters, photovoltaic cells specifically designed for near-field thermophotovoltaic devices and the experimental techniques to thoroughly test device structures. In this manuscript we report, what we believe is groundbreaking progress towards practical NFTPV devices and demonstrate unprecedented power densities of $\sim 5\text{ kW/m}^2$ beyond the blackbody limit, at vastly improved efficiency (up to 6.8%) relative to past work (\sim seven-fold improvement over the best devices reported and ~ 100 fold compared to recent work in Nature Communications: Ref. 30 of the manuscript). This was accomplished by developing novel emitter devices that can sustain temperatures as high as 1270 K and positioning them into the near-field ($<100\text{ nm}$) of custom-fabricated InGaAs-based thin film photovoltaic cells with smooth surfaces that allow for close approach between emitter and detector. In addition, we report detailed measurements of the performance of thermophotovoltaic devices across a range of emitter temperatures ($\sim 800\text{ K} - 1270\text{ K}$) and gap sizes ($70\text{ nm} - 7\text{ }\mu\text{m}$).

More specifically, an important component of the technical novelty of this work lies in the development of emitter devices that are stable to at least 200 K higher than that reported in Ref. 29 and are highly relevant to NFTPV studies. A second important advance is that we show that such heated high temperature emitters can be employed for systematically exploring NFTPV energy conversion as a function of gap size, which was not possible in the interesting work of Ref.

29. Further, we also made significant advances in preparing the PV cells by using parylene bonding techniques to transfer thin film PV cells onto a silicon substrate while maintaining excellent planarity and smoothness. As the reviewer rightly pointed out, the PV cells have significantly higher V_{oc} and fill factors than previous reports. Finally, we note that, while it is theoretically expected that increasing temperature improves the efficiency, our work represents the first report (to our knowledge) where high-power densities (~ 50 fold higher than that reported in Ref. 29 of the manuscript) and efficiencies (~ 6 fold higher than that in Ref. 29 of the manuscript) were achieved in an experimental NFTPV system operating with a heated emitter and a PV cell maintained at room temperature.

Given these advances, which are explained in detail in the manuscript and the SI, we believe that this work merits publication in Nature Communications.

Besides the above overall comment, this referee recommends the authors to address the following issues to improve scientific accuracy.

1. In general, the efficiency of a TPV system is defined as the ratio of the electrical power output to the total input power necessary for heating the emitter. On the other hand, in this paper, the authors defined the efficiency as the ratio of the electrical power output to the calculated total radiative heat transfer from the emitter to the PV cell, which neglected the graybody thermal radiation to the opposite side of the PV cell and the thermal conduction loss through the cantilever. If these losses are taken into account, the efficiency of the demonstrated device will be very low. Therefore, the authors should explicitly explain that there are two definitions of the efficiency and should calculate the efficiency of their devices according to both definitions. They should also modify the abstract and conclusion to improve scientific accuracy, for example as follows: “Here we demonstrate record power densities of $\sim 5\text{kW}/\text{m}^2$ at an estimated efficiency of 6.8 % when neglecting the far-field thermal radiation to the opposite side and the thermal conduction loss.”

The referee is correct in pointing out that the efficiency can be defined in several different ways for a near-field TPV device or system. The definition employed by us and virtually all groups (see below) in the field is naturally aimed at characterizing the performance of NFTPV device, i.e. defining the efficiency by which the radiative heat that is transferred to the NFTPV device is converted into electrical power. Therefore, we define the efficiency consistent with other TPV studies in the far-field (referred to as power conversion efficiency in Refs. 4 and 5 of the manuscript) as well as in the near-field (referred to as efficiency in Refs. 28, 29 & 30) as the ratio of the electrical power output to the total radiative heat transfer from the emitter to the PV cell. Because the emitter’s efficiency is not part of TPV system’s performance other authors have also ignored losses in the emitter due to conduction and radiation not directly incident on the PV cell. This widely adopted definition is consistent with the broadly shared vision that future NFTPV systems will be applied to recover waste heat from, for example, industrial processes. Therefore, including heat losses from the emitter that “provides or makes available” the heat that one aims to

recover is considered inappropriate to characterize a NFTPV device. Nevertheless, in order to be more accurate in our claims and as suggested by the referee, we modified our abstract and conclusion to explicitly define efficiency. Specifically, in our modified abstract we now say: “Here, we demonstrate record power densities of $\sim 5 \text{ kW/m}^2$ at an efficiency of 6.8%, where the efficiency of the system is defined as the ratio of the electrical power output generated by the PV cell to the radiative heat transfer from the emitter to the PV cell” The discussion section is modified to read: “We demonstrated efficient ($\sim 6.8\%$, excluding the heat losses through conduction and radiation from surfaces not facing the photovoltaic cell) thermophotovoltaic power generation”.

2. During the near-field experiment, did the authors perform the real-time measurement of the emitter temperature as described in Supplementary Note 6? Or, did they measure the relationship between the dissipated power and the emitter temperature (Supplementary Fig. 6) before the near-field experiment? In the latter case, was the relationship reproduced over repeated measurements, especially for high temperatures? By the way, it is counterintuitive that the temperature of 414.8°C was obtained with the heating power of 0.34 W while almost double temperature (794.1°C) was obtained with the additional heating of only 0.11 W . Why was the temperature increase of the emitter so superlinear?

We thank the reviewer for these important questions.

We clarify that the temperature measurements (as described in supplementary note 6) were performed after the near-field experiments were made on exactly the same device employed in near-field measurements. To make this point clear to future readers, we now state in the first sentence of the supplementary note 6 that: “The temperature of the emitter as a function of dissipated power was measured on the same device after the near-field experiments were performed, using a quantitative high-temperature scanning...” We note that in these experiments we directly measured the relationship between the power dissipated in the device and the temperature of the emitter at power levels identical to those used in the near-field experiments. Since the temperature rise as a function of power dissipation depends only on the temperature-dependent thermal conductivity and the radiative properties of doped silicon (conduction via residual gas molecules is negligible in both the instruments due to the high vacuum levels), our measurements of temperature are expected to accurately represent the conditions in the near-field setup. We also note that the measured temperature vs power relationship obtained using SThM was found to be repeatable in other measurements performed on a different emitter device.

With regards to the reviewer’s comment about the temperature dependence on the power dissipated, we acknowledge that there was a typographical error in the power value reported in supplementary note 6. Specifically, we incorrectly wrote 0.314 as 0.34. This error is now corrected. Next, we note that the temperature rise vs. power curve is non-linear and has varying slopes at different temperature ranges. Specifically, it has a steep rise in the temperature range of $500\text{-}800^\circ\text{C}$. At relatively low temperatures, Joule heating is volumetrically uniform in all regions of the

beams (due to relatively uniform resistivity). At higher temperatures, the resistance in locations close to the mesa increases due to reduced carrier mobility [see Ref. 3 of the supplementary information]. Thus, more heat is dissipated close to the mesa, resulting in a larger temperature rise of the mesa with a relatively small power change. This trend is reversed as the temperature further increases, because the resistance of silicon starts to drop at ~ 900 °C due to the thermal excitation of charge carriers (see Ref. 4 of supplementary information). Thus, the non-linear dependence is due to variations in the spatial distribution of Joule heat along the beams vs. temperature.

3. Related to the above comment, the upper bound of the uncertainty of the temperature measurement might be much larger than that claimed in the manuscript (10K). In the arXiv preprint of the authors (Ref. 5 in Supplementary Information), they performed the temperature calibration only at relatively low temperatures. Can the authors discuss the possible uncertainty of the temperature measurement during the near-field experiment at higher temperatures?

The uncertainty in the measured temperature of the emitter at the highest power levels is 27 K as can be seen from the data presented in supplementary Fig. 6d. In order to accurately reflect this uncertainty, we updated the efficiency estimate at the highest temperature and present it now in an uncertainty band of ± 27 K. Figures 3c, 3d and 5d were accordingly modified. We believe that this larger uncertainty in temperature measurement at the highest temperature is likely due to fluctuations in tip-sample resistance (R_{TS}) resulting from an unstable contact either due to large temperature gradients or local material degradation at the point of contact.

4. In Fig. 3, the authors show the calculated results of the electrical power and efficiency, but these values depend on not only the spectral energy transfer but also the performance of the PV cell (i.e. the open-circuit voltage and filling factor). In contrast, the short-circuit current does not depend on the latter if the quantum efficiency of the PV cell is constant. This referee recommends the authors to include the direct comparison of the measured and calculated short-circuit currents.

It is true that the electrical power and efficiency depends on the performance of the PV cell. To establish the performance of the PV cell and compare it with our model, we plotted the short-circuit current (I_{sc}) obtained at all gap sizes and temperatures as a function of open-circuit voltage (V_{oc}) in Fig. 3b. For completeness and as suggested by the referee, we now add two figures (Figs. 5a and b) showing the variation of I_{sc} and V_{oc} as functions of temperature at two gap sizes (one in the near-field and other in the far-field). Further, we have also included a new supplementary note 9, with the dark I - V response of the PV cell and the variation of the fill-factor with temperature. Agreement between measured and calculated I_{sc} is apparent in Fig. 5a. As the referee pointed out, I_{sc} depends mainly on the quantum efficiency of the PV cell and the incident photon-flux (and weakly on the series resistance). The incident flux, in-turn, depends on the temperature and emissive properties of the emitter.

Reviewer #2 (Remarks to the Author):

In this work, the authors demonstrated a near-field TPV device that shows increased power density and efficiency compared to the one some of them showed previously. This is mainly achieved by using a high-temperature doped Si emitter that has more desirable selective emission spectrum than a broadband emitter. The work shown here is novel and also has potential significance for practical applications in thermal energy conversion. As such, I think this work is suitable for publication. In the meantime, I think the following questions/points need to be addressed:

1) About the emitter. The authors used doped Si as the emitter, presumably because it is suitable for the microfabrication process used here and maybe it is stable at high temperature? However, doped Si might not be the best emitter material if the goal is to achieve narrowband emission optimized for TPV. Can the authors elaborate the reasoning behind using the doped Si (and also the doping concentration chosen here) and comment on other possible materials. Also, what about the temperature stability of this emitter? The measurement is done in UHV. Can the authors comment on the stability of the emitter if a more practical vacuum level (with higher pressure) was used.

We chose Si in our studies for a number of reasons. Importantly, Si-based emitters pave the way for future work where the effects of doping of emitters on energy conversion can be systematically studied experimentally. These Si-based emitters can also be, in future studies, readily coated with materials such as Ge, ITO, SiC and other refractory materials such as AlN or TiN. We also note that while we use Si as a broadband emitter, future work can also explore how nano-structuring the surface can lead to narrowband emission.

The referee is also correct in her/his assessment that the fabrication of silicon-based emitters is easier compared to other materials due to the suitability of Si for microfabrication. We employed highly doped Si (instead of undoped Si) to minimize the effect of temperature on the resistance, i.e. we wanted to keep the intrinsic concentration of charge carriers low such that the heating current of the emitter was dominated by contribution from the dopants over as wide a temperature range as possible. The following discussion is added in the Methods section of the manuscript to clarify this point to future readers:

“...Silicon was chosen as the emitter material due to its ease of micro-fabrication and compatibility with other thin-film material growth processes such as ITO, TiN, AlN etc. The heavily doped ($\sim 3 \times 10^{19} \text{ cm}^{-3}$ B-doped) Si emitter allowed us to reliably heat the emitter in a large temperature range...”

Finally, we note that our measurements were performed at a vacuum level of $\sim 1 \mu\text{Torr}$. In our experiments the emitter was first equilibrated at a temperature of 1170 K for ~ 12 h and then the temperature was raised to 1270 K for ~ 3 h during the measurement. No structural degradation of

the surfaces or change in the performance of the emitters was noticed after these experiments. Further, additional experiments were carried out to heat the emitters to ~ 1200 K at vacuum levels of a few mTorr for 2-3 hour durations. These experimental trials also showed no degradation of the devices. To implement practical devices, future studies would be needed to understand the long-term stability of Si emitters, with various protective coatings, under a range of pressures. These points are now discussed in the discussion section of the manuscript, where we say: "...Future studies on the long-term stability of the emitters at high temperatures with various protective coatings, under a range of pressures could enable realization of practical devices..."

2) About the heat loss. The authors determined the efficiency using the measured power output from the PV device and the calculated radiative heat transfer flux, which is relevant for this device geometry. Can the authors also discuss the overall energy efficiency (electrical output divided by electrical power input or joule heating applied on the emitter). More specifically, I think it will be useful to quantify the heat loss (in terms of the percentages) through different pathways. One large heat loss I could imagine is through the side walls of the Si emitter (Si is much thicker compared to the near field gap).

While the overall electrical-to-electrical conversion efficiency is not relevant to the goal of our study and NFTPV experiments in general, we quantify the heat loss through different pathways as the referee suggested. The major pathway for heat loss from the emitter is through conduction, while heat is also lost from the backside and sidewalls of the emitter through radiation to the surroundings. Thus, when dissipating 549 mW (power dissipation corresponding to $T_{\text{emitter}} = 1270$ K) of electrical power in the beams, 2 mW (0.4%) is lost as heat through radiation from surfaces facing the PV cell, and ~ 10 mW (1.8%) as radiation from the backside and the side walls of the emitter. Therefore, the major heat loss is through conductance along the beams of 537 mW (98%). Of the 2 mW radiative heat transfer to the PV cell, 143 μ W is extracted as electrical power output leading to the $\sim 6.8\%$ efficiency reported in this study. The overall conversion efficiency from electrical power (dissipated in the emitter) to electrical power (output in the PV cell) is 0.026%. Electrical-to-electrical power conversion efficiencies of the emitter and NFTPV system at various temperatures and gap sizes can be estimated by comparing Fig. 3c and supplementary Fig. 6d. This analysis of the different pathways by which heat is dissipated from the emitter has now also been added in the third paragraph of the Supplementary note 3.

3) About the SBG photons. The authors mentioned that the SBG energy transfer is enhanced with the near field. I guess this is related to the RHT flux (as shown in fig. 4a). This leads to the drop in the efficiency in the near field. However, isn't the TPV cell capable of reflecting (and thus recycling) the SBG photons? So the flux of the SBG photons is presumably not important here. I guess this is the key idea behind refs. 4 & 5 (for far field TPV). Likewise and related to point 2 above, I am not sure if the efficiency calculation should include the RHT flux of the SBG photons. If these photons are recycled, then it should not be included in the calculation (as done in ref. 4).

If they are not efficiently recycled, then perhaps the authors should explain this in detail.

We thank the referee for this comment. As we pointed out in the manuscript, employing a thin-film PV cell with a back-surface reflector (BSR) can reduce the SBG absorption by reflecting low-energy photons for recycling to the emitter. While significant recycling of low-energy photons is observed in the near-field (below the black-body limit), residual absorption in the gold reflector results in some SBG absorption as seen in red curve (100 nm gap size) of Fig. 4a. To further clarify this point, we have now added supplementary note 10, discussing the improvement in comparison with a thick PV cell without proper recycling of SBG photons. The large SBG energy transfer for the case of a thick PV cell (red line in Supplementary Fig. S11c) indicates poor recycling of SBG photons, while in a thin-film PV cell and Au BSR, a significant reduction of the SBG energy transfer is observed (blue line in Supplementary Fig. S11c). We note that, in principle, by employing an air-bridge reflector, the reflectance can be further improved leading to efficiencies of up to 18% at $T_{\text{emitter}} = 1270$ K. Although such air-bridge devices are conceptually simple, implementing them in a NFTPV system requires overcoming several technical challenges, as discussed in the manuscript.

In addition to adding the supplementary note 10, we have also modified the last few paragraphs of the manuscript (marked in yellow) to clarify the discussion regarding SBG reflection. Specifically, we now say:

“...For example, at a gap size of 100 nm, significant enhancement over the blackbody limit (black dashed line) can be seen in the ABG energy transfer, while considerable suppression of SBG energy transfer below the blackbody limit is observed, due to the incorporation of a thin-film back reflector (see Supplementary Note 10 for comparison with a bulk PV cell). The residual SBG energy transfer has contributions from surface phonon-polaritons in the low frequency range ($\sim 14\%$ of Q_{RHT} in 0.0124 - 0.073 eV range) while the rest of the absorption primarily occurs in the Au BSR ($\sim 55\%$ of Q_{RHT} in 0.074 - 0.74 eV range)...”

“...The efficiencies in the NF are slightly smaller than in the far-field, owing to absorption in the Au film reflector, which can be mitigated by engineering the devices to further suppress SBG energy transfer. This can be achieved by employing an air-gap PV cell which has recently been shown to support very efficient SBG suppression⁴. Such devices must be engineered to address a host of technical requirements (smooth surfaces, planarity, temperature compatibility) before they can be adapted for NFTPV studies...”

Reviewer #3 (Remarks to the Author):

This research is focusing on energy conversion from thermal energy to electricity using near-field radiation transfer. The nanometer-sized parallel gap between a doped-Si emitter and an InGaAs thermophotovoltaic (TPV) cell surfaces is rigorously and successfully established in a vacuum chamber using the smallest effective area to confirm the view factor of almost unity. The output power was enhanced by near-field effect by a factor of 8, compared with that by far-field radiation transfer. Also, numerical results agree well with the experimental ones. As a result, the manuscript is acceptable for publication.

However, Reviewer would like to give some comments as follows.

(1) The author estimated the conversion efficiency from the measured output power and the calculated input energy from the emitter to the TPV cell. However, in Figure 3d, each symbol indicates 'Exp' which seems a conversion efficiency obtained from only experiment. Reviewer suggests that those symbol data should be removed. Generally speaking, the conversion efficiency should be defined using the input energy measured by experiment and the output power obtained by experiment. As the authors know well, it is not easy to measure the input energy experimentally. All researchers agree with you. In your research, fortunately, the calculated output power agrees well with the measured one. If so, the author had better show the conversion efficiency obtained from only calculation which are the shaded region in Fig.3d. Otherwise, it is not fair to make a comparison between your results and previous results obtained only from experiments.

We thank the reviewer for these comments. We agree with the suggestion of modifying the efficiency figures to include only the bands, where the efficiency is defined as the ratio of the measured power output to the calculated radiative heat transfer in the corresponding temperature intervals. Accordingly, changes were made to figures 3d and 5d and in the corresponding text.

(2) In order to keep the TPV cell temperature at 298 K, the author should set up a cooling system at the bottom of the Si block substrate. The reader can't confirm how to make the thermal management from Figure 1a. In addition, in the Supplementary Figure 5a, the bottom of the Si block substrate is kept at constant (maybe at 298K) in numerical simulation. The volume of the Si block (400 x 400 x 200 microns) is very small; as a result, the heat capacity is also very small. It means if the authors could not prepare the cooling system at the bottom of the Si block, the temperature of cell with Si block would rise up easily by an input heat of 2mW within a short time. The author should mention clearly how to keep the cell temperature at almost room temperature for all experiments. Is there any cooling system or another large substrate contacted with the Si block?

We agree with the referee that it is not obvious from Fig. 1a how the PV cell is mounted. In our experiments, the PV cell and the emitter are mounted in a custom-built nanopositioner. To clarify

this point to future readers we now include in Supplementary note 5 a one-dimensional thermal network analysis (Supplementary Fig. S5a) and a schematic of the bottom assembly onto which the PV cell is mounted. This section now reads as follows:

“Supplementary Note 5. Thermal modelling to determine temperature rise of the PV cell

In our experiments, the PV cell is mounted on the bottom assembly of our custom-built nanopositioner as shown in Supplementary Fig. 5a. As can be seen, the PV cell substrate ($10 \times 8 \times 0.5$ mm, red square) is mounted on a DIP chip carrier (shown in yellow, ‘B’) with silver paste which in-turn is attached to an aluminum platform. From a simple one-dimensional thermal model, we estimate the thermal conductances from point ‘A’ to point ‘F’ (a large heat-sink at ambient temperature). The smallest thermal conductance is between point ‘A’ to point ‘B’. In other words, the thermal resistance is dominated by the structure shown in Supplementary Fig. 5b. In order to identify the temperature rise of the PV cell due to the heat transfer from the emitter, we then performed FEM analysis, solving the thermal conduction equation for a representative PV cell when it is illuminated by the hot emitter. Supplementary Fig. 2b shows the simulated structure consisting of a thick silicon substrate ($0.4 \times 0.4 \times 0.2$ mm). Thin layers of parylene ($2 \mu\text{m}$), gold (400 nm), GaAs ($1.5 \mu\text{m}$) and gold ($1 \mu\text{m}$) are included to represent a realistic model of the PV cell consistent with the description in supplementary note 2 above. The bottom surface is maintained at a constant temperature of 300 K, while heat (Q_{input}) is applied on the top surface of the PV cell at its center. The corresponding temperature rise (ΔT_{PV}) is calculated for different inputs of heat (Supplementary Fig. 2a). The estimated radiative heat transfer (Q_{RHT}) to the PV cell at the highest temperature (1270 K) achieved in this work and a gap of 100 nm, is ~ 2 mW. Thus, based on this analysis the temperature rises of the PV cell in the experiments performed in this study are expected to be less than 1 K. We note that in the real experiments, the PV cell may be heated to a slightly higher temperature (~ 5 K), possibly due to heat emission from the beams of the emitter which is not considered in the calculation of Q_{RHT} and interfacial thermal resistances in the multilayers of the PV cell that are also not included in the simulation.”

Supplementary Figure 5. Thermal management of PV cell. *a*, Schematic of the bottom assembly of the nanopositioner onto which the PV cell is mounted. Typical thermal conductances along different sections of this assembly are shown. *b*, Boundary conditions applied for FEA of the temperature distribution of the PV cell due to heat transfer from the mesa of the emitter to the PV cell. *c*, Temperature increase at point 'A' (Supplementary Fig. 5a) at the center of the top surface of the PV cell, as a function of heat input to the PV cell.

REVIEWERS' COMMENTS

Reviewer #1 (Remarks to the Author):

In the revised version of the manuscript, the authors have addressed all the comments of the three referees, thereby improving the scientific accuracy of the manuscript. This referee agrees with the authors that the demonstrations made in this work are important progress towards practical NFTPV systems. Therefore, the manuscript is now almost suitable for publication in Nature Communications. As the last comment, this referee recommends the authors to briefly point out the similarities (e.g. Si emitter and InGaAs-based thin film cell) between this work and Ref. 29 in the manuscript, not just compare the system performance between them.

Reviewer #2 (Remarks to the Author):

The authors have satisfactorily addressed my comments regarding the heat loss and SBG photon recycling. I think the revised work shown here is suitable for publication in Nature Communications.

Reviewer #3 (Remarks to the Author):

Thank you for your improvement of manuscript.
I becomes better.

RESPONSE LETTER

We once again thank the referees for taking the time to review our manuscript and for providing valuable feedback to improve the manuscript.

Reviewer #1 (Remarks to the Author):

In the revised version of the manuscript, the authors have addressed all the comments of the three referees, thereby improving the scientific accuracy of the manuscript. This referee agrees with the authors that the demonstrations made in this work are important progress towards practical NFTPV systems. Therefore, the manuscript is now almost suitable for publication in Nature Communications. As the last comment, this referee recommends the authors to briefly point out the similarities (e.g. Si emitter and InGaAs-based thin film cell) between this work and Ref. 29 in the manuscript, not just compare the system performance between them.

We thank the referee for the feedback. Following her/his recommendation we have included the requested information. The relevant text now reads as follows:

“Nevertheless, all of these demonstrations show limited efficiency and power density, with the best-reported device²⁹ (using a Si emitter and an InGaAs cell) featuring a maximum efficiency of ~0.98% at a power density of ~120 W/m² when operated at a maximum temperature of 1040 K.”

Reviewer #2 (Remarks to the Author):

The authors have satisfactorily addressed my comments regarding the heat loss and SBG photon recycling. I think the revised work shown here is suitable for publication in Nature Communications.

We thank the referee for recommending our work for publication.

Reviewer #3 (Remarks to the Author):

Thank you for your improvement of manuscript. It becomes better.

We thank the referee for acknowledging the improvements in our manuscript.